# Secretory Factors from Calcium-Sensing Receptor-Activated SW872 Pre-Adipocytes Induce Cellular Senescence and A Mitochondrial Fragmentation-Mediated Inflammatory Response in HepG2 Cells

**DOI:** 10.3390/ijms24065217

**Published:** 2023-03-09

**Authors:** Lautaro Briones-Suarez, Mariana Cifuentes, Roberto Bravo-Sagua

**Affiliations:** 1Laboratory of Obesity and Metabolism (OMEGA), Institute of Nutrition and Food Technology (INTA), University of Chile, Santiago 7830490, Chile; 2Department of Nutrition and Public Health, Faculty of Health and Food Sciences, University of Bío-Bío, Chillán 3800708, Chile; 3Center for Exercise, Metabolism and Cancer (CEMC), Faculty of Medicine, University of Chile, Santiago 8380453, Chile; 4Advanced Center for Chronic Diseases (ACCDiS), Santiago 8380492, Chile; 5Interuniversity Center for Healthy Aging (CIES), Consortium of Universities of the State of Chile (CUECH), Santiago 8320216, Chile

**Keywords:** calcium-sensing receptor, cell senescence, pre-adipocyte, hepatocyte

## Abstract

Adipose tissue inflammation in obesity has a deleterious impact on organs such as the liver, ultimately leading to their dysfunction. We have previously shown that activation of the calcium-sensing receptor (CaSR) in pre-adipocytes induces TNF-α and IL-1β expression and secretion; however, it is unknown whether these factors promote hepatocyte alterations, particularly promoting cell senescence and/or mitochondrial dysfunction. We generated conditioned medium (CM) from the pre-adipocyte cell line SW872 treated with either vehicle (CM_veh_) or the CaSR activator cinacalcet 2 µM (CM_cin_), in the absence or presence of the CaSR inhibitor calhex 231 10 µM (CM_cin+cal_). HepG2 cells were cultured with these CM for 120 h and then assessed for cell senescence and mitochondrial dysfunction. CM_cin_-treated cells showed increased SA-β-GAL staining, which was absent in TNF-α- and IL-1β-depleted CM. Compared to CM_veh_, CM_cin_ arrested cell cycle, increased *IL-1β* and *CCL2* mRNA, and induced p16 and p53 senescence markers, which was prevented by CM_cin+cal_. Crucial proteins for mitochondrial function, PGC-1α and OPA1, were decreased with CM_cin_ treatment, concomitant with fragmentation of the mitochondrial network and decreased mitochondrial transmembrane potential. We conclude that pro-inflammatory cytokines TNF-α and IL-1β secreted by SW872 cells after CaSR activation promote cell senescence and mitochondrial dysfunction, which is mediated by mitochondrial fragmentation in HepG2 cells and whose effects were reversed with Mdivi-1. This investigation provides new evidence about the deleterious CaSR-induced communication between pre-adipocytes and liver cells, incorporating the mechanisms involved in cellular senescence.

## 1. Introduction

Cell senescence is described as a mechanism whereby a dividing cell enters a stable cell cycle arrest upon a stressing stimulus while remaining metabolically active. Senescent cells show the senescence-associated secretory phenotype (SASP) which involves the production of pro-inflammatory cytokines and other signaling factors and they become unresponsive to mitogenic and apoptotic signals [1]. Depending on the context, cell senescence can result in beneficial or detrimental effects to the organism. In young individuals or upon acute damage, senescence-related cell cycle arrest and SASP contribute to tumor suppression, wound healing, and tissue homeostasis [2].

Obesity is described as a chronic low-grade inflammatory process, where hypertrophic adipocytes accumulate in adipose tissue. This organ has an important endocrine role, secreting pro- or anti-inflammatory factors such as tumor necrosis factor α (TNF-α) and interleukin-1β (IL-1β) or interleukin-10 (IL-10) and adiponectin, respectively. The pro-inflammatory cytokines are associated with dysfunctional adipose tissue, which alters local and systemic metabolism, leading to obesity-related comorbidities [3,4]. Adipose tissue dysfunction has also been related to age-related diseases [3].

Adipose progenitor cells or “pre-adipocytes” are a cell population within adipose tissue with the ability to proliferate and differentiate into mature adipocytes, maintaining homeostasis through tissue plasticity and expandability [5,6,7]. An inflammatory environment can alter the homeostasis of pre-adipocytes, reducing their differentiation capacity and increasing their proliferation and pro-inflammatory cytokine expression [8,9]. Multiple inflammatory pathways have been described in adipose tissue, including the activation of the calcium-sensing receptor (CaSR), a G-protein coupled receptor that responds to several ligands and signals through multiple pathways. Pre-adipocytes express CaSR, and its activity is involved in the release of pro-inflammatory cytokines TNF-α and IL-1β [10,11,12], which may promote stress-mediated cellular senescence development in other tissues [13,14].

Aging and obesity modulate the cellular senescence program, leading to the accumulation of senescent cells that promote tissue dysfunction, chronic inflammation, and fibrosis in the liver [4,15]. Alterations in liver cell function are important because of its role in the control of metabolism, biomolecule secretion, and energy homeostasis [16,17]. The liver is rich in mitochondria and is responsible for 15% of the total body oxygen consumption. Depending on the nutrient demand, the liver can oxidize various nutrients, such as carbohydrates and lipids, which are metabolized in mitochondria. Indeed, the liver can provide glucose to other tissues and, during fasting, its mitochondria can oxidize the fatty acids released from adipose tissue, producing ketones and ATP. Therefore, mitochondria are key players for hepatocyte energy homeostasis and their dysfunction is involved in the development of metabolic diseases [18,19].

It has been suggested that cell senescence and mitochondrial dysfunction are key hallmarks of aging [20]. During cell senescence, dysfunctional mitochondria accumulate due to a reduction in mitophagy [21]. Moreover, the senescent phenotype in hepatocytes is associated with decreased lipid oxidation, which favors hepatic inflammation and lipid accumulation (steatosis) [22].

The inflammatory environment generated by cytokines such as IL-1β and TNF-α alters the liver’s morphology and function, promotes fibrosis, mitochondrial dysfunction, and increases the number of senescent cells [22,23,24]. However, it is unknown whether the secretion products of pre-adipocytes after CaSR activation promote cell senescence and mitochondrial dysfunction in hepatocytes. Our aim was to investigate if CaSR activation induces pro-inflammatory secretion in the human pre-adipocyte cell line SW872, leading to pro-inflammatory cytokine expression, cellular senescence, and mitochondrial dysfunction in HepG2 cells.

## 2. Results

### 2.1. Pro-Inflammatory Secretion Factors from CaSR-Activated SW872 Pre-Adipocytes Induce Cell Senescence in HepG2 Cells

To evaluate the pro-senescence effect of pre-adipocyte-secreted factors, we obtained conditioned medium (CM) from SW872 cells treated with the CaSR activator cinacalcet 2 µM for 16 h (CM_cin_). As the control, we used CM from SW782 cells treated with vehicle alone (CM_veh_). The senescence-associated β-galactosidase (SA-β-GAL) assay (a marker of residual lysosomal activity associated with cell senescence) [13,25] showed that CM_cin_-treated HepG2 cells displayed higher SA-β-GAL expression, as well as cytomegaly compared to CM_veh_-treated cells (*p* < 0.05) (Figure 1a). We quantified SA-β-GAL expression both as the count of positive cells and colorimetry (Figure 1b,c). To test whether these changes depended on CaSR activation, we obtained CM from SW872 cells treated with cinacalcet in the presence of the CaSR inhibitor calhex 231 10 µM (CM_cin+cal_), which abolished the effect of CM_cin_ (Figure 1b,c).

To further characterize the senescent phenotype in our model, we evaluated the p53/p21/p16 signaling cascade. Cell cycle regulation by p53, p21, and p16 can be activated by stress signals, such as DNA damage or mitochondrial dysfunction [26,27,28]. CM_cin_-treated HepG2 cells showed increased protein levels of p16 and p21, but not p53, as assessed by Western blot analysis (*p* < 0.05) (Figure 2a–d). Again, treatment with CM_cin+cal_ failed to induce any change, thereby supporting the role of CaSR activation in this process.

Cell cycle arrest is a hallmark of cell senescence, which we evaluated by the location of the protein Ki67. This protein is diminished in the nucleus of senescent cells, and together with other markers, allows us to assess cell senescence [29]. As expected, HepG2 cells treated with CM_cin_ showed a decreased Ki67 immunofluorescence signal compared with the control condition (*p* < 0.05). This change did not take place in the CM_cin+cal_ condition (Figure 3a,b).

Another key feature of senescent cells is the senescence-associated secretory phenotype (SASP), which consists of the induction of pro-inflammatory cytokines and chemoattractants, such as IL-1β and CCL2, respectively [30,31]. Our results show that HepG2 cells treated with CM_cin_ increased the relative abundance of both IL-1β and CCL2 mRNA compared to CM_veh_ (*p* < 0.05), while CM_cin+cal_ prevented these changes (Figure 4a–c).

To explore the mediators of this pro-senescence effect, we separately immunoprecipitated IL-6, IL-1β, or TNF-α from the SW872-derived CM_cin_, given the evidence of their release by preadipocytes [10,11,32] in addition to their association with stress-associated cell senescence [33,34,35]. Our approach specifically reduced the levels of each cytokine from the CM (Figure 5a). We verified the effectiveness of the procedure by observing that each cytokine was enriched in the pellet of their respective immune complex (Figure 5b). Immunoprecipitation of either TNF-α (CM_TNF-α[−]_) or IL-1β (CM_IL-1β[−]_) prevented the increase in SA-β-GAL in CM_cin_-treated HepG2 cells, while IL-6 immunoprecipitation (CM_IL-6[−]_) did not evoke a significant effect (Figure 5c–e).

### 2.2. Secretion Products from CaSR-Activated SW872 Pre-Adipocytes Alter the Mitochondrial Dynamics and Function of HepG2 Cells

Given that cell senescence is characterized by a general decrease in cell function including mitochondrial dysfunction [27,36], we evaluated key factors for mitochondrial dynamics. Our results show that the protein levels of OPA1, which maintains mitochondrial cristae structure, and PGC-1α, which promotes mitochondrial biogenesis, significantly decrease in HepG2 cells exposed to CM_cin_ (*p* < 0.05) compared to CM_veh_. In the case of MFN2 protein levels, which mediates mitochondrial fusion, there was a trend to decrease (*p* < 0.08). Conversely, the protein levels of DRP1, which participates in mitochondrial fission, increased upon treatment with CM_cin_ (Figure 6a–f). As with previous results, these observations were reversed in the CM_cin+cal_ condition.

Next, we analyzed the mitochondrial transmembrane potential (Δψ) through immunofluorescence using the Δψ-sensitive probe MitoTracker Orange (MTO) normalized by mtHsp70 as a marker of mitochondrial mass. We observed that HepG2 cells exposed to CM_cin_ had a decreased MTO/mtHsp70 fluorescence ratio (*p* < 0.05), indicative of decreased Δψ (Figure 7a,b). In addition, CM_cin_-treated cells presented a higher number of mitochondria per cell compared to the CM_veh_ condition and a trend towards reduced average mitochondrial area (*p* < 0.08) (Figure 7c,d). Altogether, these results indicate that CM_cin_ treatment in pre-adipocytes induces the secretion of factors that promote mitochondrial fragmentation and the decrease of bioenergetics in HepG2 cells, thus suggesting mitochondrial dysfunction.

### 2.3. Inhibition of Mitochondrial Fission in HepG2 Cells Prevents Cell Senescence Induced by Cinacalcet-Treated SW872 Pre-Adipocyte Secretion Factors

We addressed whether the observed mitochondrial fragmentation contributes to the development of MC_cin_-induced cell senescence in HepG2 cells by inhibiting mitochondrial network fragmentation using the DRP1 inhibitor Mdivi-1 during the last 24 h of the treatment with SW872 CM. As expected, Mdivi-1 treatment prevented CM_cin_-induced mitochondrial fragmentation, maintaining the mitochondrial number similar to CM_veh_-treated cells (Figure 8a,c,d). Similarly, Mdivi-1 treatment prevented the decrease in Δψ observed upon CM_cin_ treatment (Figure 8b). 

We next investigated how Mdivi1 affects the mitochondrial bioenergetics of HepG2 treated with CM_cin_. We observed that while CM_cin_ did not significantly alter baseline respiration compared to CM_veh_, it decreased both non-ATP associated and maximal capacity respiration rates, confirming mitochondrial dysfunction. In both cases, treatment with Mdivi-1 reverted said changes (Figure 9a–c).

Finally, we assessed whether mitochondrial fragmentation inhibition counteracts CM_cin_-induced cell senescence, evaluated as cell cycle arrest and SASP in HepG2 cells. Indeed, Mdivi-1 treatment prevented the decrease in Ki67 nuclear fluorescence (Figure 10a,b) induced by CM_cin_, as well as the increase in IL-1β and CCL2 mRNA levels (Figure 11a,b). Altogether, these results highlight the importance of mitochondrial fragmentation for the progression of cell senescence induced by CaSR-activated pre-adipocytes.

## 3. Discussion

The present work evaluated whether CaSR activation in SW872 pre-adipocytes promotes pro-inflammatory signaling that induces senescence and mitochondrial dysfunction in HepG2 cells. Our observations indicate that CaSR activation in the SW872 cell line stimulated the production of factors that increased senescence, the production of pro-inflammatory cytokines, cell cycle arrest, and mitochondrial dysfunction markers in HepG2 cells. 

In pre-adipocytes, CaSR activation has been shown to promote the secretion of the pro-inflammatory cytokines TNF-α and IL-1β [10,11]. Elevated circulating TNF-α is associated with an increase in cellular senescence markers, such as SA-β-GAL, p21, and p53 in the liver and kidneys [37]. Our observations are consistent with other investigations where TNF-α was able to increase SA-β-GAL [37]. Moreover, we also observed that removing IL-1β prevented an increase in SA-β-GAL. Thus, our results highlight that both IL-1β and TNF-α in the conditioned medium from CaSR-activated pre-adipocytes contribute to cellular senescence development in the HepG2 cell line.

Studies in macrophages have shown that exposure to conditioned medium from senescent pre-adipocytes increase p16 and p21 expression, which was attributed to the secretory products that make up the SASP, such as TNF-α and CCL2 [38,39]. Other reports on human vascular smooth muscle cell cultures have shown that IL-1β exposure increases p16 and p21 expression through a mechanism involving SIRT1 deregulation [40]. Decreased activation of the SIRT1 metabolic pathway has been observed in models of aging and inflammation. In rats, SIRT1 activation promoted mitochondrial biogenesis and the endogenous antioxidant system through PGC1a and NRF2 activity on neuronal tissue [41]. A study in cells from the nucleus pulposus showed that TNF-α promoted the cellular senescence markers SA-β-GAL and p21, a change that was related to greater activity of NF-κB [42]. Unlike p16 and p21, in our experiments we did not observe significant changes on p53, a protein that modulates p21 expression. In relation to this, it has been described that p21 can be regulated in a p53-independent manner through ERK1/2 and p38 MAPK [43].

It is well-accepted that the inflammatory activity of TNF-α and IL-1β initiates stress-driven cellular senescence. The process begins with an increase in cellular senescence markers such as SA-β-GAL and greater expression of pro-inflammatory cytokines and chemoattractants such as IL-1β and CCL2 [44]. In this context, the increased expression of IL-1β and CCL2 triggers the spread of inflammation and the attraction of immune cells, which eliminate senescent cells [14,44,45].

Studies in animal models have shown that the low expression of OPA1 is related to the increase in the pro-inflammatory cytokines TNF-α and IL-1β in the circulation [46]. In the liver of OPA1-deficient mice, the markers for cellular senescence, SA-β-GAL and p21, were increased [46]. We observed low expression of PGC-1α in CM_cin_-treated HepG2 cells, which is consistent with results in mouse liver tissue showing that stimuli such as a high-fat diet decreases PGC-1α expression and increases cellular senescence markers, such as SASP [47]. Moreover, the low expression of PGC-1α is related to reduced protection against oxidative stress and cellular senescence [48]. In the CM_cin+cal_ treatment, MFN2 recovered its expression and DRP1 decreased compared to CM_cin_. Low expression of MFN2 has been associated with increased fragmentation of the mitochondrial network and lower membrane potential [49,50]. It has been reported that pro-inflammatory stimuli such as TNF-α increase mitochondrial dysfunction markers such as the fragmentation of the mitochondrial network [51].

In senescent cells, mitochondrial alterations such as fragmentation in the mitochondrial network or loss in the Δψ are common features [52,53]. Our results coincide with those reported in colon and melanocyte cell lines exposed to overexpression of TNF-α, which decreased the Δψ and increased the fragmentation of the mitochondrial network. After an early process of apoptosis, the remaining cells show mitochondrial dysfunction and increased cytokine expression, which is associated with cellular senescence [53]. The evidence on altered mitochondrial dynamics and cellular senescence is under discussion because the results may vary depending on the cell type under study [22,54]. Based on our findings, we propose that the secretion products in CM_cin_ affect mitochondrial morphology through decreased expression of proteins that promote organelle biogenesis and fusion, as well as increased fragmentation of the mitochondrial network. In addition, treated cells presented an alteration in mitochondrial bioenergetics, showing a lower Δψ and lower respiratory rate. These data suggest that CM_cin_ damages the mitochondrial network, and thus limits the biogenesis and bioenergetics of this organelle.

Losing mitochondrial network integrity favors the development of mitochondrial dysfunction, and in our HepG2 cells treated with CM_cin_ this indicator increased. In experiments in HepG2 cells where Ki67 was pharmacologically reduced, Ki67 was recovered after Mdivi-1 treatment [55]. Furthermore, in a rat model of ischemia, Mdivi-1 increased Δψ and the authors attribute this behavior to the regulatory action of the reagent on mitophagy mediated by PINK/Parking proteins [56]. It was also shown in LPS-exposed cardiomyocytes that Mdivi-1 contributes to the recovery of Δψ [57]. The effects of Mdivi-1 on the Δψ regulation would be related to mitochondrial membrane integrity maintenance. Our results show that Mdivi-1 inhibited the increase in IL-1β and CCL2 expression caused by CM_cin_. Published studies in animal models exposed to amyloid β reveal that treatment with Mdivi-1 protects hippocampal cells from increased expression of IL-1β, which changes in mitochondrial dynamics, decreasing the MFN2 and OPA1 proteins [58]. In a cell model, it has been observed that treatment with Mdivi-1 before LPS exposure impaired the increase in pro-inflammatory cytokines such as IL-1β and CCL2 [59]. 

The present work addresses a line of research that innovates on the relationship between CaSR activation in pre-adipocytes and the development of the senescent phenotype and mitochondrial dysfunction in hepatic cells. It highlights and supports previous research on the importance of pre-adipocytes as potential promoters of deleterious effects in other cell types, further evidencing the participation of mitochondrial dysfunction as a regulator of the senescent phenotype. Future studies should also include a wider use of CaSR inhibition strategies such as the CaSR-negative modulator calhex 231 and possibly others, and furthermore with CaSR gene downregulation via siRNA, in order to better understand the involvement of this receptor in the observed effects. Further characterization of the senescent phenotype would have been desirable, such as cell cycle analysis through flow cytometry or chromatin remodeling or DNA damage via fluorescence microscopy. In addition, we did not explore other functional outcomes in senescent HepG2 cells, such as lipid uptake or insulin signaling. Finally, this model of cellular communication can be used with other cell types. Thus, the effects of conditioned media from CaSR-activated pre-adipocytes can be assessed in smooth muscle or endothelial cells as a future research direction. 

Our results indicate that inhibiting mitochondrial network fragmentation protects HepG2 cells from changes related to CM_cin_ exposure, seen in the recovery of Δψ and the decreased expression of pro-inflammatory cytokines. In summary, the conditioned medium from pre-adipocytes after CaSR activation promotes senescence and mitochondrial dysfunction in hepatocytes. Our investigation provides evidence about the communication between pre-adipocytes and liver cells mediated by CaSR activation, raising new questions such as further pre-adipocyte conditioned medium characterization or the consequences of CaSR activation at the organismal level on liver senescence phenotype and disease.

## 4. Materials and Methods

### 4.1. Cell Culture

A human liposarcoma-derived SW872 pre-adipose cell line (HTB-92, ATCC, Manassas, VA, USA) was grown in DMEM/F12 medium. A human hepatocellular carcinoma-derived HepG2 cell line (HB-8065, ATCC, Manassas, VA, USA) was cultured in MEM medium. All media were supplemented with 10% fetal bovine serum and antibiotics (penicillin–streptomycin) and the cells were maintained in a 37 °C 5% CO_2_ atmosphere. The medium was changed for a fresh medium twice per week.

### 4.2. SW872 Cells’ Conditioned Media Collection

SW872 cells were cultured at a 10,000 cells/cm^2^ density in 100 mm plastic culture dishes. To obtain conditioned media, the cells were treated with either vehicle (DMSO) or 2 µM cinacalcet with or without 30 min of pre-incubation with the negative allosteric CaSR modulator calhex-231 (10 µM). After 16 h, the media were replaced with fresh DMEM/F12 and conditioned for 24 h. The conditioned media were centrifuged at 800× *g* for 10 min at 4 °C and stored at −80 °C until use.

### 4.3. HepG2 Cells’ Exposure to Conditioned Media and Mdivi1

HepG2 cells were cultured at 9000 cells/cm^2^ density in culture dishes, according to each experimental protocol. The cells were washed twice with PBS and the media were replaced with fresh MEM media containing 50% of the conditioned media from SW872 cells. The HepG2 cells were then grown for 5 days under standard culture conditions, replacing the media for fresh conditioned media after 3 days.

For treatment with Mdivi1, the cells were conditioned as described above, and during the last 24 h of exposure they were treated with vehicle (DMSO) of 50 µM of Mdivi1.

### 4.4. Immunoprecipitation

IL-6, IL-1β, or TNF-α were immunoprecipitated from the CM_cin_. For that, 10 μg of IL-6, IL-1β, or TNF-α antibodies (Table 1) were incubated with 200 μL of hydrated Protein A-Sepharose CL-4B (17-0963-03, Sigma, St. Louis, MO, USA) for 16 h to obtain a solution with the immobilized antibody. Then, the immobilized antibody solution was mixed with 1 mL of CM_cin_ 12 h at 4 °C in a rotary mixer. The sample was then centrifuged at 3000× *g* for 5 min at 4 °C. Supernatants were used to treat HepG2 cells and evaluate SA-β-GAL. To analyze cytokine protein levels in the immunoprecipitate, the pellets were resuspended in 30 µL of Laemmli loading buffer and evaluated through Western blot analysis.

### 4.5. HepG2 SA-β-GAL Activity

The cells were cultured in 96-well plates and observed under an inverted phase contrast microscope before and after treatment for 5 days. Images were recorded with a Motic AE2000 camera and Motic Images Plus 2.0 ML software (Motic, Vancouver, BC, Canada). The cultures were treated with the senescence-associated β-galactosidase staining kit (Cell Signaling Technology, Danvers, MA, USA) following the manufacturer’s instructions. After incubating overnight at 37 °C, the cells were examined by microscopy and the respective blue staining in positive cells was recorded. Next, the number of positive cells was normalized by the total number of cells in the photograph to obtain an indicator of the percentage of senescent cells in each experiment. Then, X-gal blue products were dissolved with DMSO and the absorbance was measured at 630 mn [60].

### 4.6. RNA Isolation, Reverse Transcription, and mRNA Expression by RT-PCR

The cells were cultured in 6-well plates and after experimentation, Trizol (Invitrogen, Life Technologies, Carlsbad, CA, USA) was used to lyse the HepG2 cells and isolate total RNA following the manufacturer’s instructions. RNA was reverse-transcribed into complementary DNA (cDNA) using a high-capacity cDNA reverse transcription kit (Applied Biosystems, Foster City, CA, USA). The cDNA levels were analyzed with a step-one real-time PCR system using the SYBR FAST qPCR kit (Applied Biosystems, Foster City, CA, USA) and specific primers for: *il-1β* (forward: 5′ GGACAAGCTGAGGAAGATGC 3′; reverse: 5′ TCGTTATCCCATGTGTCGAA 3′, NM_000576), *ccl2* (forward: 5′ TGTCCCAAAGAAGCTGTGATCT 3′; Reverse: 5′ GGAATCCTGAACCCACTTCTG 3′; NM_002982), and *gapdh* (forward: 5′ GAAGGTGAAGGTCGGAGTCAAC 3′; reverse: 5′ CAGAGTTAAAAGCAGCCCTGGT 3′; NM_020). Thermal cycling consisted of an initial pre-incubation cycle of 20 s at 95 °C, followed by 40 cycles of 30 s at 95 °C. The results were normalized for the *gapdh* gene and relative mRNA levels were calculated using the Pfaffl method [61].

### 4.7. Protein Levels by Western Blot Analysis

The cells were cultured in 6-well plates and after experimentation, they were washed three times with cold PBS and lysed with NP40 buffer (Invitrogen™, Carlsbad, CA, USA), complete protease inhibitor (#11697498001, Sigma-Aldrich, St. Louis, MO, USA), and PhosSTOP phosphatase inhibitor (#4906845001, Sigma-Aldrich, St. Louis, MO, USA). The lysed cells were centrifuged at 12,000× *g* for 15 min and the supernatant was stored at −80 °C until use. At concentrations of 1 μg/μL, the proteins were subjected to electrophoresis in SDS-polyacrylamide gels under denaturing conditions and were electrotransferred to a 0.2 μm pore PVDF membrane. The membranes were then blocked with TBS 5% skim milk 0.05% Tween 20 (#7949, Sigma, St. Louis, MO, USA). 

The following proteins were detected after 16 h of incubation with the primary antibodies: p16, p21, p53, PGC1α, MFN2, DRP1, OPA1, and β-actin (Table 1). The detection of immune complexes was performed with the incubation of anti-rabbit IgG, anti-mouse IgG, or anti-goat IgG peroxidase-conjugated secondary antibodies, depending on the origin of each primary antibody (Table 2). The membranes were incubated for 1 min with chemiluminescence reagents (#20-500-500A and 20-500-500B, Biological Industries, Cromwell, CT, USA), the digitized luminescent signal was detected in a LI-COR C-Digit scanner 3600 (LI-COR Biosciences, Lincoln, NE, USA), and the intensity of the bands was quantified with the ImageJ program (National Institutes of Health, USA). The expression of each protein was normalized by β-actin and expressed as arbitrary units.

### 4.8. Immunofluorescence

The HepG2 cells were seeded in 12-well plates with 0.17 mm and treated as indicated. For analysis of mitochondrial membrane potential, the cells were loaded with MitoTracker Orange 400 nM (M7510, Invitrogen™, Carlsbad, CA, USA) for 25 min at 37 °C. Then, the culture medium was removed and the cells were washed twice with PBS. The cells were fixed with 4% paraformaldehyde at 4 °C for 15 min. The fixation medium was removed and washed twice with cold PBS. To permeabilize the cells, they were incubated with PBS 0.1% Triton X-100 for 10 min. After permeabilization, the cells were washed twice with cold PBS and blocked with PBS 3% BSA for 1 h at room temperature. The samples were treated with primary antibodies in PBS 3% BSA overnight at 4 °C in a humid chamber protected from light.

The primary antibodies and their dilutions were: anti-Ki-67 (#9449, Cell Signaling, proliferation marker) 1:400 or anti-mtHsp70 (MA3-028, Thermo Fisher Scientific, mitochondrial marker) 1:500, according to the assay. The next day, the coverslips were washed twice with cold PBS to remove excess primary antibodies, and the samples were incubated for 1 h with an Alexa Fluor 488 secondary antibody (A32723, Thermo Fisher Scientific, Waltham, MA, USA) 1:600 in a chamber protected from light at room temperature. After washing with PBS, the samples were incubated with the fluorescent probe Hoechst (B2261, Merck (Rahway, NJ, USA), nucleus marker) 1:1000 diluted in PBS BSA 3% for 15 min at room temperature in a chamber protected from light. The coverslips were then washed twice with cold PBS to remove excess probes and mounted on a clean slide with Dako fluorescence mounting medium (S3023, Dako-Agilent, Santa Clara, CA, USA). The samples were stored at 4 °C protected from light until analysis.

### 4.9. Image Capture and Processing

Images were taken with a Zeiss LSM-5, Pascal 5 Axiovert 200 confocal microscope, with a Plan-Apochromat 40×/1.5 Oil DIC objective, using 365 (for Hoechst), 488 (for Alexa Fluor 488), and 555 nm (for MitoTracker Orange) excitation lasers. For each independent experiment, it averaged the signal from 5 to 15 cells. The analysis was performed on 1 focal plane corresponding to the equator of the cells. The pixel size was 68 nm, following the Nyquist sampling criterion. The images thus obtained were deconvolved, the background noise was subtracted, a median filter was applied, and fluorescence intensity was quantified using the ImageJ software.

### 4.10. Oxygraphy

HepG2 cells were seeded in 60 mm dishes. After the 5 days of treatment, the cells were trypsinized and resuspended in a Clark electrode chamber (Strathkelvin Instruments, North Lanarkshire, Scotland) to quantify the oxygen consumption of living cells. Basal respiration was measured for 5 min at 25 °C, then 10 μg/mL of oligomycin was added to assess non-ATP-associated respiration for 5 min, followed by 200 nM CCCP to quantify maximal respiration capacity for another 5 min. Then, the cells were recovered to quantify total proteins using a BCA kit to normalize oxygen consumption rates.

### 4.11. Statistical Analysis

Data are shown as the mean ± the standard error of the mean (SEM). Differences between conditions were evaluated with the non-parametric Friedman test and multiple comparisons were made with the Dunn’s test. The Wilcoxon signed rank test was used to detect differences in the results obtained through real-time PCR. A *p* value less than 0.05 was considered as a significant change, while *p* values less than 0.1 were considered a trend.

## Figures and Tables

**Figure 1 ijms-24-05217-f001:**
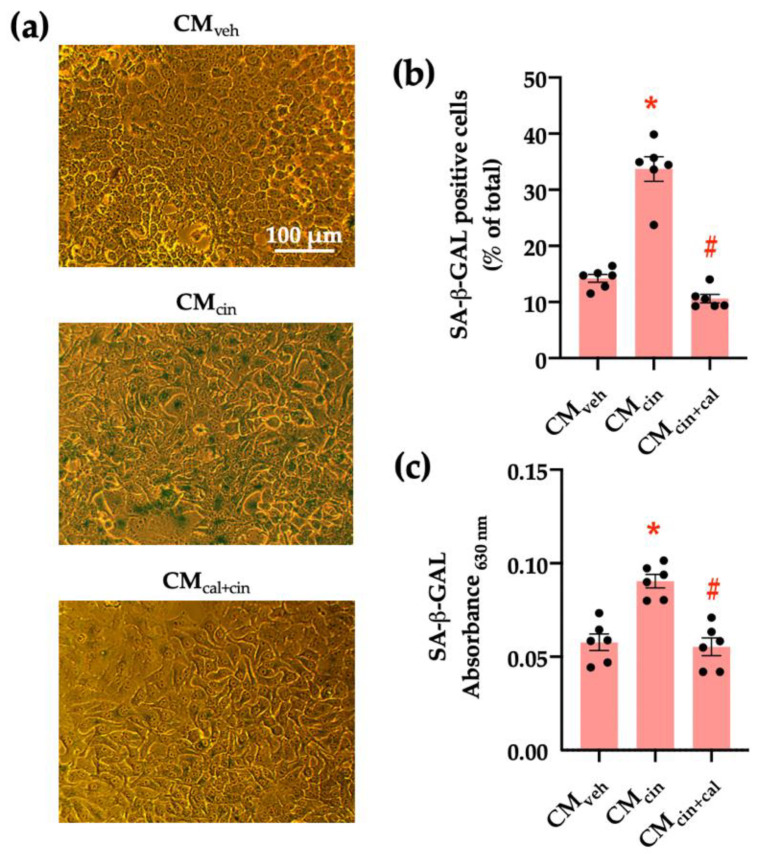
CaSR activation in SW872 pre-adipocytes induces the secretion of factors that induce cell senescence in HepG2 hepatocytes: (**a**) Representative images of SA-β-GAL staining of HepG2 cells exposed to the CM_veh_, CM_cin_, and CM_cin+cal_ (scale bar = 100 μm); (**b**) Quantification of SA-β-GAL positive cells normalized by total cells in the field; and (**c**) SA-β-GAL absorbance recorded at 630 nm. Each dot represents an individual experiment (*n* = 6). * *p* < 0.05 versus the control, ^#^
*p* < 0.05 versus CM_cin_, Friedman’s test, and Dunn’s post-hoc test.

**Figure 2 ijms-24-05217-f002:**
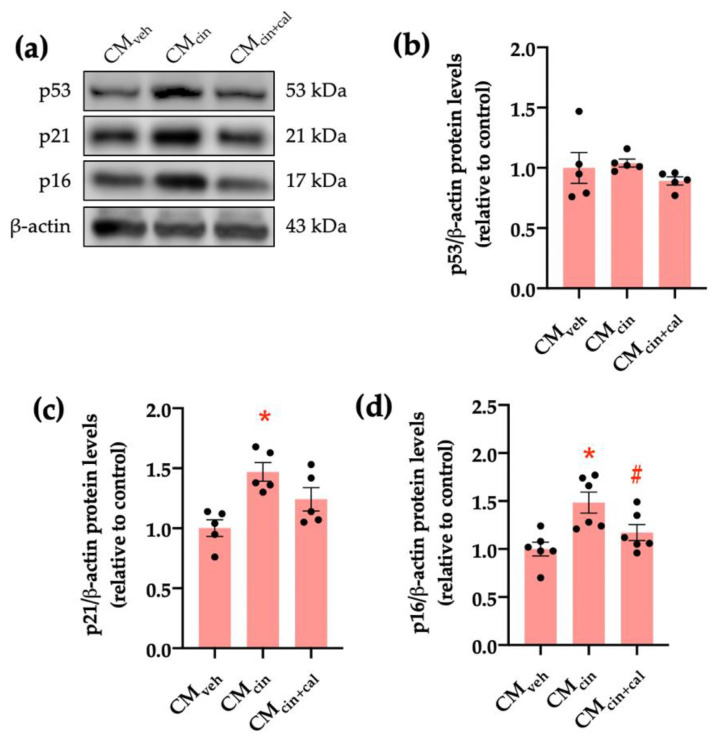
CaSR activation in SW872 pre-adipocytes induces the secretion of factors that activate the p16/p21/p53 cascade in HepG2 cells: (**a**) immunodetection of p53, p21, and p16 protein levels in HepG2 cells exposed to the CM_veh_, CM_cin_, and CM_cin+cal_; (**b**) quantification of p53 protein levels; (**c**) quantification of p21 protein levels; (**d**) quantification of p16 protein levels. Each dot represents an individual experiment *(n* = 5–6). * *p* < 0.05, versus the control, ^#^
*p* < 0.05 versus CM_cin_, Friedman’s test, and Dunn’s post-hoc test.

**Figure 3 ijms-24-05217-f003:**
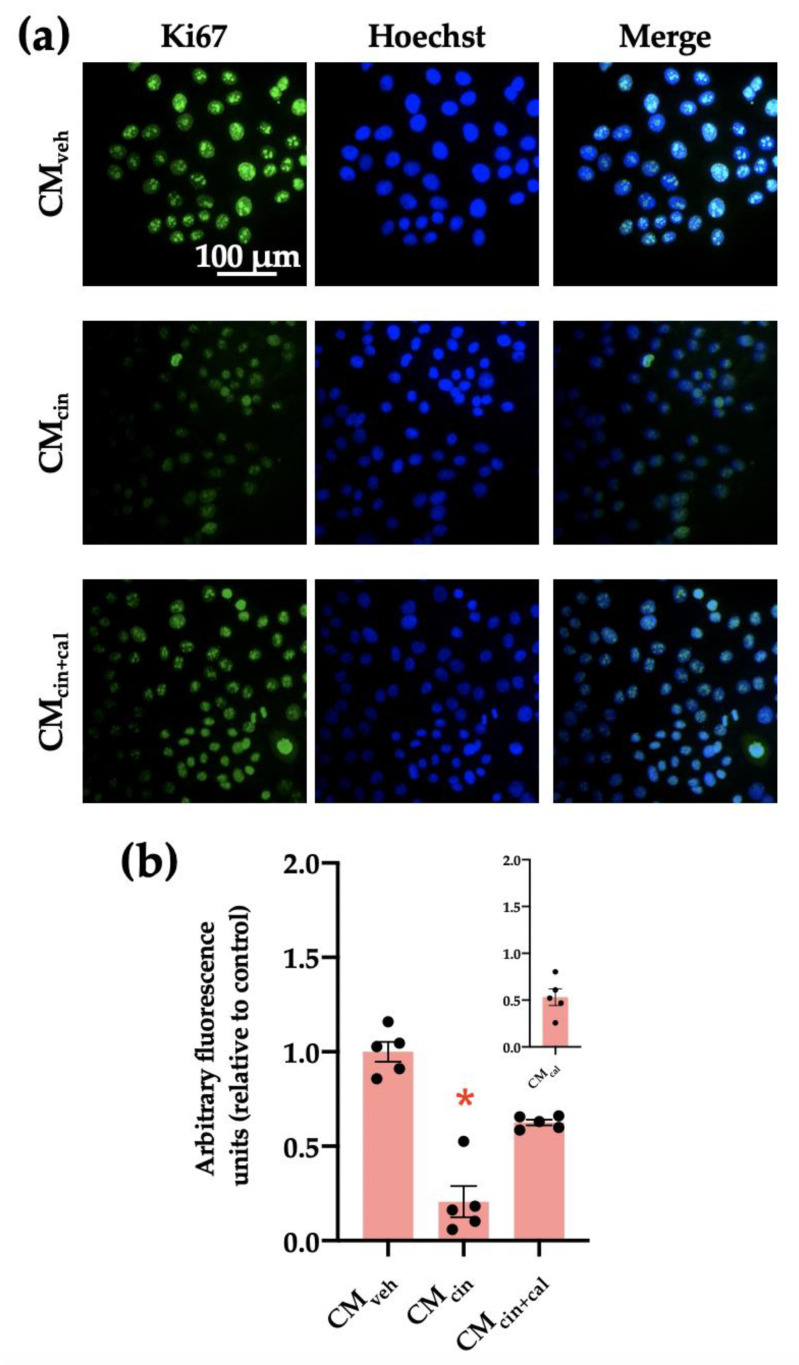
CaSR activation in SW872 pre-adipocytes induces the secretion of factors that induce cell cycle arrest in HepG2 cells: (**a**) representative images of HepG2 cells exposed to the CM_veh_, CM_cin_, and CM_cin+cal_. Hoechst (blue) was used to identify the nuclei and anti-Ki67 antibody (green) to evaluate the nuclear localization of Ki67 (scale bar = 100 μm); (**b**) quantification of total Ki67 immunofluorescence. The inset shows CM_cal_ immunofluorescence relative to the control. Each dot represents an individual experiment (*n* = 5). * *p* < 0.05 versus the control, Friedman’s test, and Dunn’s post-hoc test.

**Figure 4 ijms-24-05217-f004:**
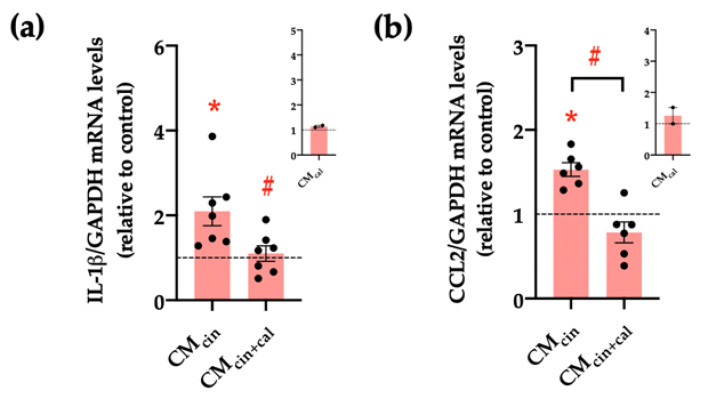
CaSR activation in SW872 pre-adipocytes induces the secretion of factors that induce the secretion of pro-inflammatory cytokines in HepG2 cells: relative abundance of (**a**) *IL-1β* and (**b**) *CCL2* mRNA in HepG2 cells exposed to CM_cin_ and CM_cin+cal_; expressed as fold from CM_veh_, (represented by the dotted line). mRNA levels were normalized to *GAPDH* mRNA. The insets show CM_cal_ normalized mRNA levels relative to the control. Each dot represents an individual experiment (*n* = 2–7). * *p* < 0.05 versus the control, ^#^
*p* < 0.05 versus CM_cin_, and the Wilcoxon signed rank test.

**Figure 5 ijms-24-05217-f005:**
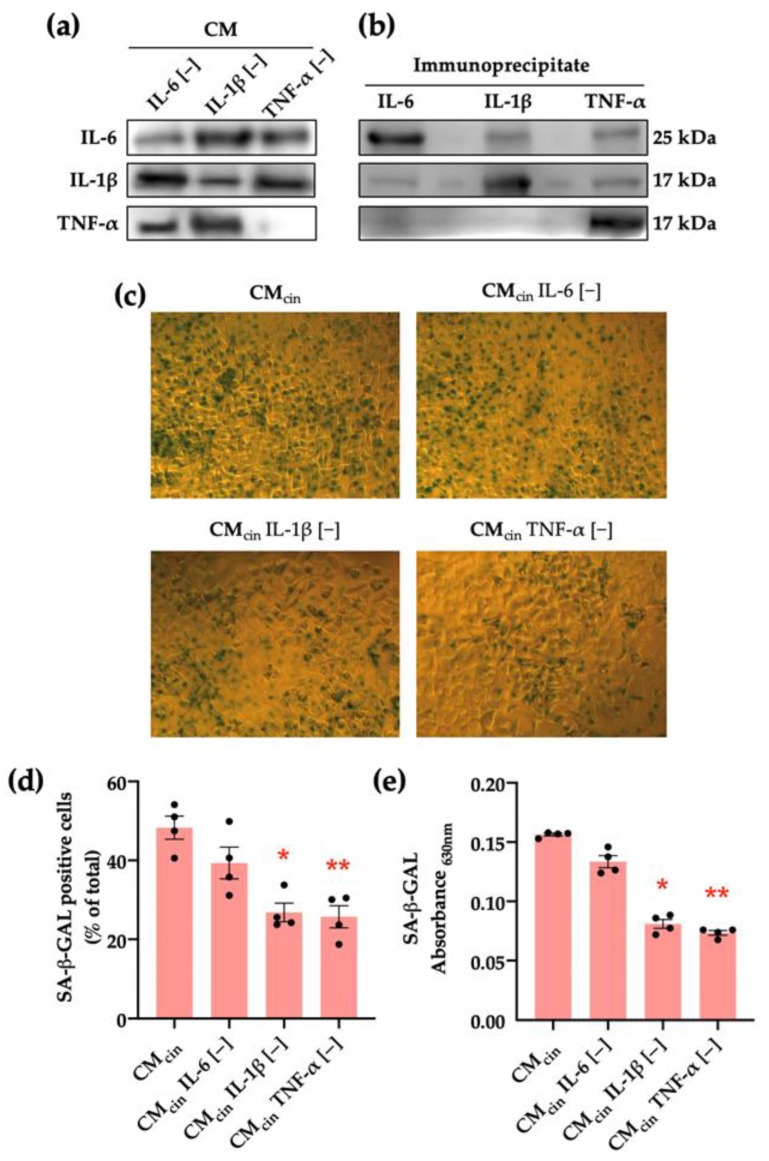
CaSR activation in SW872 pre-adipocytes elevates IL-1β and TNF-α secretion, which induce cell senescence in HepG2 cells: (**a**) immunodetection of IL-6, IL-1β, and TNF-α protein levels in CM_cin_ after immunoprecipitation (*n* = 1); (**b**) immunodetection of IL-6, IL-1β, and TNF-α protein levels in the CM_cin_ pellet after immunoprecipitation (*n* = 1); (**c**) Representative images of SA-β-GAL staining of HepG2 cells exposed to CM_cin_, CM_IL-6 [−]_, CM_IL-1β [−]_, and CM_TNF-α [−]_ (scale bar = 100 μm); (**d**) quantification of SA-β-GAL-positive cells normalized by total cells in the field; (**e**) SA-β-GAL absorbance recorded at 630 nm. Each dot represents an individual experiment (*n* = 6). * *p* < 0.05, and ** *p* < 0.01, the versus control, Friedman’s test, and Dunn’s post-hoc test.

**Figure 6 ijms-24-05217-f006:**
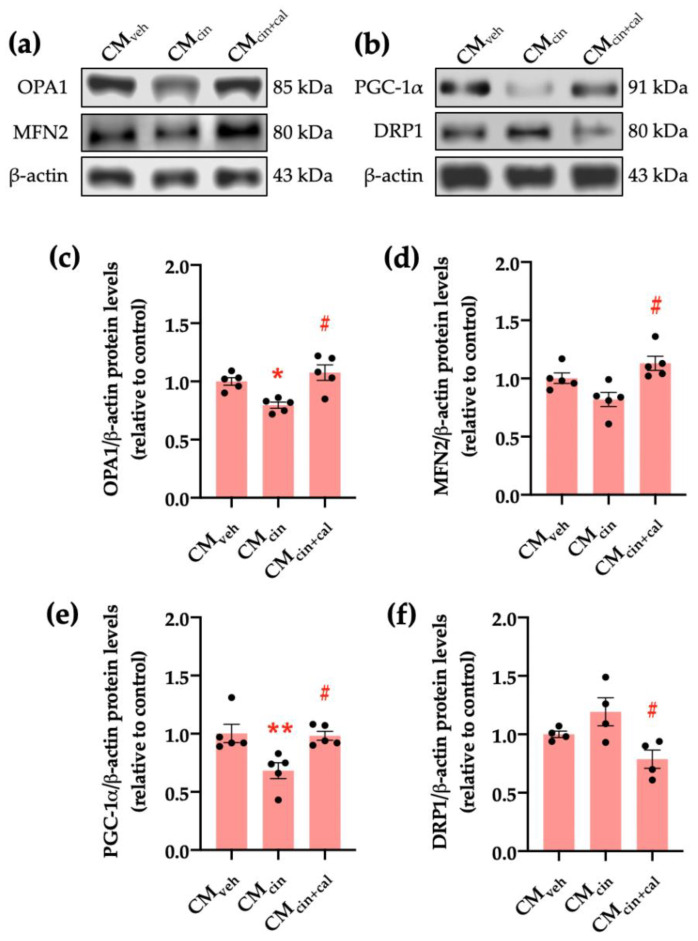
CaSR activation in SW872 pre-adipocytes induces the secretion of factors that alter proteins governing mitochondrial dynamics in HepG2 cells: (**a**) immunodetection of OPA1 and MFN2 protein levels in HepG2 cells exposed to CM_veh_, CM_cin_, and CM_cin+cal_; (**b**) immunodetection of PGC-1α and DRP1 protein levels in treated HepG2 cells; quantification of (**c**) OPA1; (**d**) MFN2; (**e**) PGC-1α; and (**f**) DRP1 protein levels. Each dot represents an individual experiment (*n* = 4–5). * *p* < 0.05 and ** *p* < 0.01 versus the control, ^#^
*p* < 0.05 versus CM_cin_, Friedman’s test, and Dunn’s post-hoc test.

**Figure 7 ijms-24-05217-f007:**
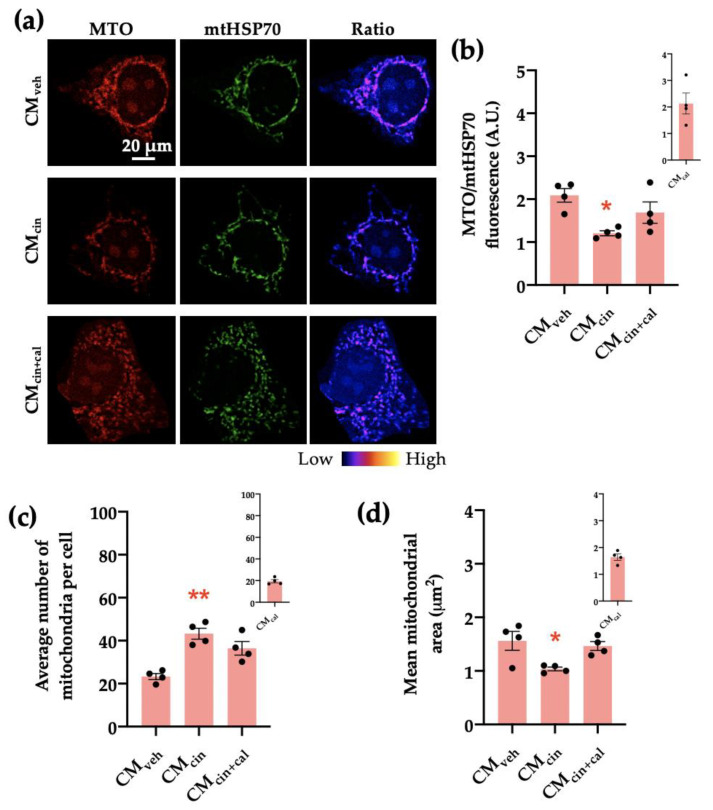
CaSR activation in SW872 pre-adipocytes induces the secretion of factors that induce mitochondrial alterations in HepG2 cells: (**a**) representative immunofluorescence images obtained through confocal microscopy of HepG2 cells treated with CM_veh_, CM_cin_, or CM_cin+cal_ stained with MitoTracker Orange (MTO, red) and an anti-mtHsp70 antibody (green); the ratio of MTO/mtHSP70 fluorescence is shown in pseudo colors (right column of images); (**b**) quantification of the overall fluorescence ratios between MTO mtHsp70; (**c**) quantification of the mean mitochondrial area; (**d**) quantification of the average number of mitochondria per cell. The insets show CM_cal_ levels relative to the control. Each dot represents an individual experiment (*n* = 4). * *p* < 0.05 and ** *p* < 0.01 versus the control, Friedman’s test, and Dunn’s post-hoc test.

**Figure 8 ijms-24-05217-f008:**
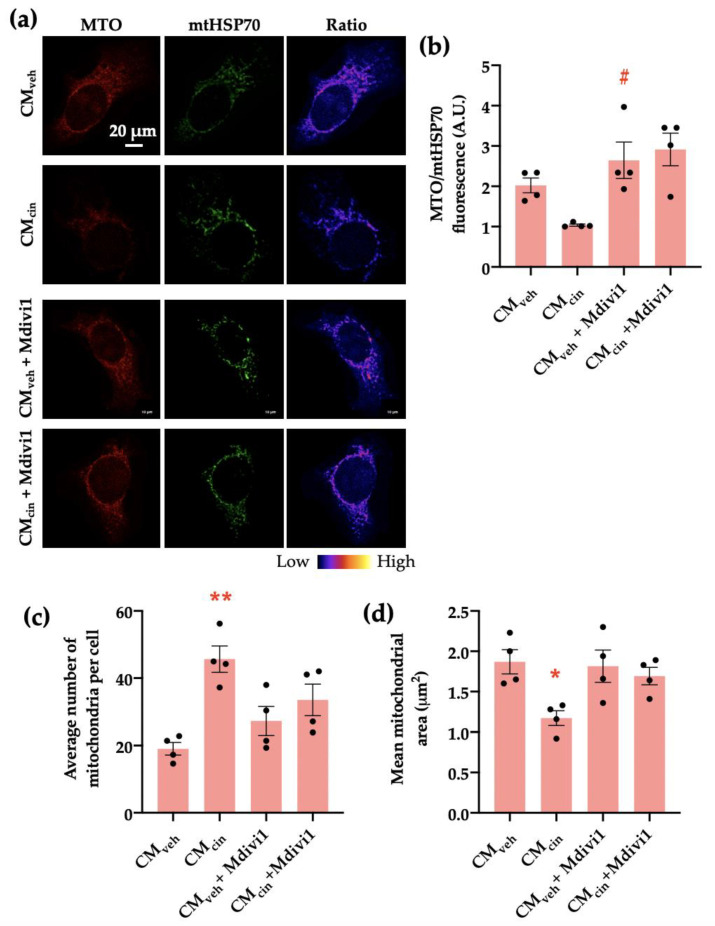
Mdivi1 prevents mitochondrial alterations induced by CM_cin_ on HepG2 cells: (**a**) representative immunofluorescence images obtained through confocal microscopy of HepG2 cells treated with CM_veh_ and CM_cin_ and subsequently treated or not with Mdivi-1, stained with MitoTracker Orange (MTO, red) and an anti-mtHsp70 antibody (green); the ratio of MTO/mtHSP70 fluorescence is shown in pseudo colors (right column of images); (**b**) quantification of the overall fluorescence ratios between MTO mtHsp70; (**c**) quantification of the mean mitochondrial area; (**d**) quantification of the average number of mitochondria per cell. Each dot represents an individual experiment (*n* = 4). * *p* < 0.05 and ** *p* < 0.01 versus the control, ^#^
*p* < 0.05 versus CM_cin_, Friedman’s test, and Dunn’s post-hoc test.

**Figure 9 ijms-24-05217-f009:**
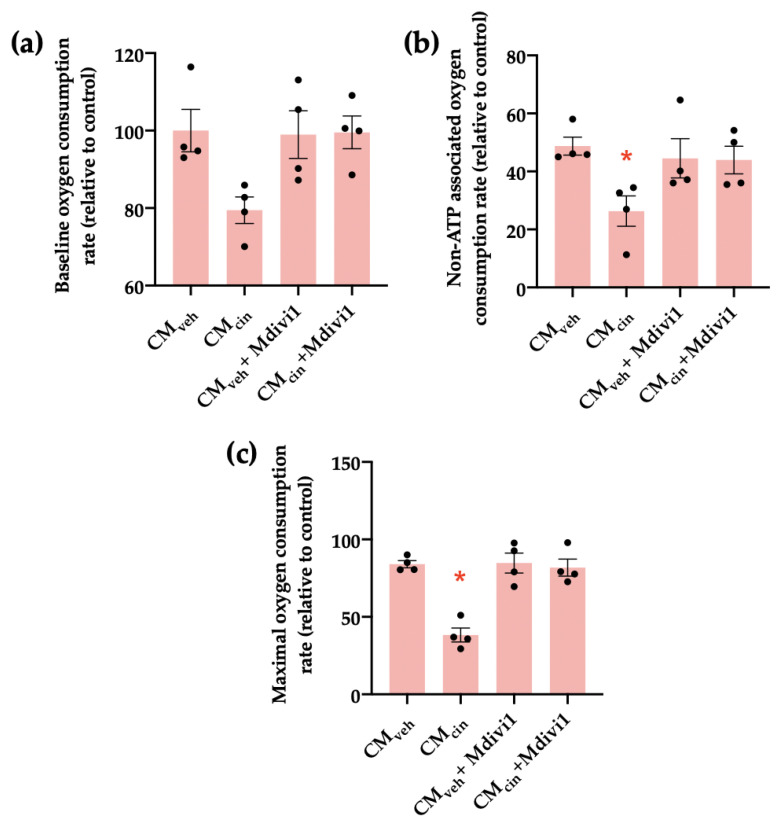
Mdivi1 inhibits the mitochondrial bioenergetics alterations induced by CM_cin_ on HepG2 cells: (**a**) quantification of baseline oxygen consumption rates of HepG2 cells treated with CM_veh_ or CM_cin_ and subsequently treated or not with Mdivi-1; (**b**) quantification of non-ATP-associated oxygen consumption rates of treated HepG2 cells; (**c**) quantification of maximal capacity oxygen consumption rates of treated HepG2 cells. Each dot represents an individual experiment (*n* = 4). * *p* < 0.05 versus the control, Friedman’s test, and Dunn’s post-hoc test.

**Figure 10 ijms-24-05217-f010:**
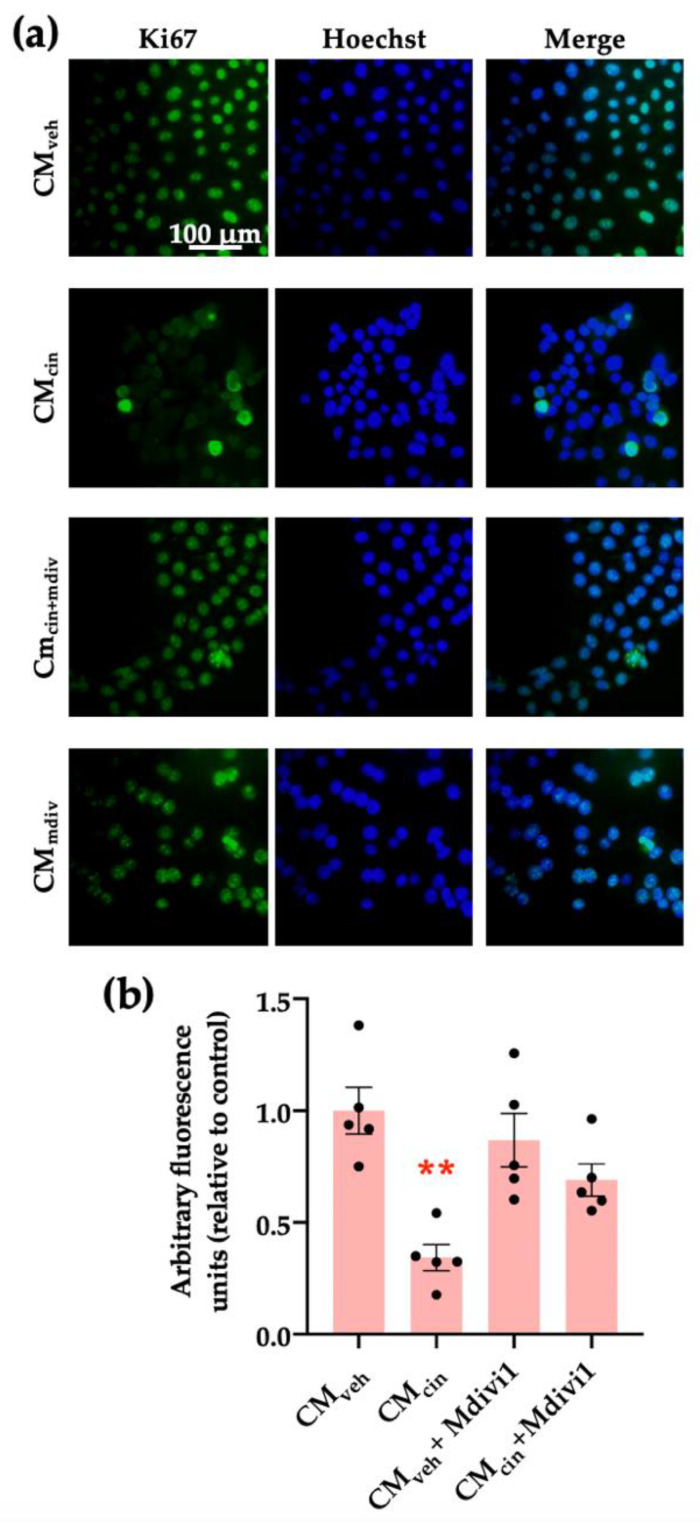
Inhibition of mitochondrial fission prevents the cell cycle arrest induced by CM_cin_ in HepG2 cells: (**a**) representative images of HepG2 cells exposed to the CM_veh_, CM_cin_, CM_veh_ + Mdivi1, and CM_cin_ + Mdivi1. Hoechst (blue) was used to identify the nuclei and anti-Ki67 antibody (green) to evaluate the nuclear localization of Ki67 (scale bar = 100 μm); (**b**) quantification of total Ki67 immunofluorescence. Each dot represents an individual experiment (*n* = 5). ** *p* < 0.01 versus the control, Friedman’s test, and Dunn’s post-hoc test.

**Figure 11 ijms-24-05217-f011:**
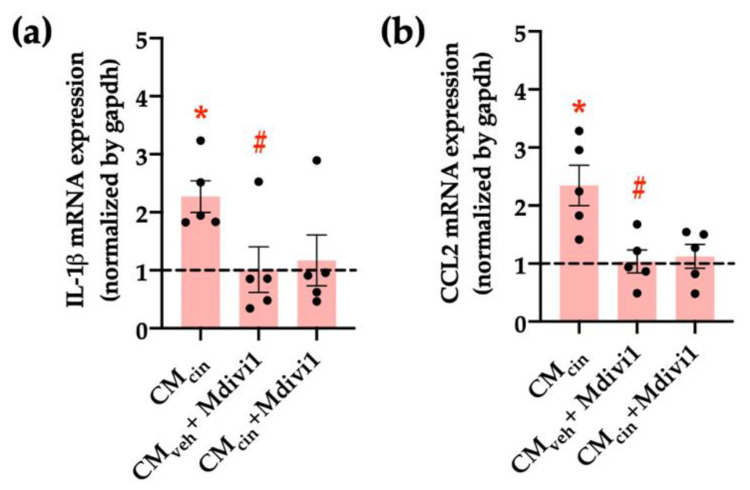
Inhibition of mitochondrial fission prevents the increase in pro-inflammatory cytokines induced by CMcin in HepG2 cells: relative abundance of (**a**) *IL-1β* and (**b**) *CCL2* mRNA in HepG2 cells exposed to CM_cin_, CM_veh_ + Mdivi1, and CM_cin_ + Mdivi1, expressed as fold from CM_veh_, represented by the dotted line. mRNA levels were normalized to *GAPDH* mRNA. Each dot represents an individual experiment (*n* = 5). * *p* < 0.01 versus the control, ^#^
*p* < 0.05 versus CM_cin_, and the Wilcoxon signed rank test.

**Table 1 ijms-24-05217-t001:** Primary antibodies for Western blot analysis.

Antibody	Code	Manufacturer
**Cell cycle markers**		
p16	ab108349	Abcam (Cambridge, UK)
p21	ab109520	Abcam (Cambridge, UK)
p53	#48818	Cell Signaling Technology (Danvers, MA, USA)
**Pro-inflammatory cytokines**		
TNF-α	sc-1350	Santa Cruz Biotechnology (Dallas, TX, USA)
IL-1β	sc-7884	Santa Cruz Biotechnology (Dallas, TX, USA)
IL-6	sc-283443	Santa Cruz Biotechnology (Dallas, TX, USA)
**Mitochondrial dynamics**		
PGC-1α	NBP1-04676	Novus Biologicals (Centennial, CO, USA)
MNF2	#9482	Cell Signaling Technology (Danvers, MA, USA)
DRP1	sc-271583	Santa Cruz Biotechnology (Dallas, TX, USA)
OPA1	ab157457	Abcam (Cambridge, UK)
**Housekeeping**		
β-actin	sc-47778	Santa Cruz Biotechnology (Dallas, TX, USA)

**Table 2 ijms-24-05217-t002:** Secondary antibodies for Western blot analysis.

Antibody	Code	Manufacturer
Mouse IgG	sc-516102	Santa Cruz (Dallas, TX, USA)
Rabbit IgG	sc-2357	Santa Cruz (Dallas, TX, USA)
Goat IgG	sc-2028	Santa Cruz (Dallas, TX, USA)

## Data Availability

The data presented in this study are available on request from the corresponding author.

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
