# Peer review of "Secretory Factors from Calcium-Sensing Receptor-Activated SW872 Pre-Adipocytes Induce Cellular Senescence and A Mitochondrial Fragmentation-Mediated Inflammatory Response in HepG2 Cells"

_ijms, 2023, doi:10.3390/ijms24065217_

Round 1

Reviewer 1 Report

The manuscript by Briones et al. describes an interesting crosstalk between adipocytes and hepatocytes, fuelled by CaSR. Although the presented results are interesting, some questions/concerns remain before this manuscript can be considered for publication.

Major

-                  Please also show significancy level comparing CMcin with CMcin+cal in all figures.

-    Inclusion of an CMcal condition would be necessary for all figures/experiments to see whether inhibition of CaSR at basal conditions also has effects on HepG2 cells.

-              In general for the graphs in which CMcal is already included, its basal effects do not really match the expectation. How can this be explained? Does CMcal show opposite effects on SW872 cells in comparison to CMcin (in other words; is confirmed that the inhibition works properly in basal conditions?)

-          Regarding Figure 3. CMcal has a big effect on the Ki67 levels and seems to also downregulate its nuclear expression. How can this be explained, as an upregulation would be expected.

-          Regarding figure 4, the n=2 for the CMcal condition is underpowered and n-numbers should be increased. Furthermore, ELISA experiments should be performed to investigate whether HepG2 cells also secrete more cytokines. For this CM medium should be changed again to ensure the cytokines are deriving from HepG2 cells and not the SW872 cells. (Similarly these experiments should be performed for Figure 11).

-          This brings me to the next point, whether the observed effects are transient or long-lasting. If the CM is removed from the HepG2 cells, how long can the key observations then still be observed?

-          Do the authors also observe other functional effects on HepG2 cells, like e.g. changes in lipid uptake capacity?

-          The authors describe an interesting phenomenom by which CaSR in pre-adipocytes induce senescence in author cells. Is this specific for hepatocytes or are comparable effects also seen in other cell types, like e.g. endothelial cells or smooth muscle cells. For this, selected key experiments should be replicated in one of these cell types.

Minor

-          Please double figure 7D, this seems to be the incorrect graph (and copy of figure 8D).

-          In Figure 9A all control values are 100. Please adjust the normalization so that also the variability in the control group can be assessed/visualized.

Author Response

Major

  1. Please also show significancy level comparing CMcin with CMcin+cal in all figures.

We marked all the differences versus CMcin with “#” when they were significant and added the explanation in the legends (figures 1-2, 4, 6, 8, 11).

  1. Inclusion of an CMcal condition would be necessary for all figures/experiments to see whether inhibition of CaSR at basal conditions also has effects on HepG2 cells.

We agree with the reviewer that this is a limitation of our study. Unfortunately, the production of conditioned media is a long and expensive process and thereby is the limiting reagent in our study design, which involved a single batch of each CM in order to minimize this as a source of variability. We produced and used CMcal with very little excess, thinking only about key experiments and further experiments would require creating another batch that would not really be comparable with the other experiments the way we originally designed it. We regret we cannot perform the suggested experiments, but acknowledging their relevance, we have included it as one of the limitations of our study (lines 333-336).

  1. In general, for the graphs in which CMcal is already included, its basal effects do not really match the expectation. How can this be explained? Does CMcal show opposite effects on SW872 cells in comparison to CMcin (in other words; is confirmed that the inhibition works properly in basal conditions?)

In our experiments, CMcin promoted senescence as well as mitochondrial dysfunction. CMcal, in turn, did not have a basal effect per se. Our explanation for that observation is that when CaSR was activated by cinacalcet in preadipocytes, they released into the CM those factors that induce the senescent state in HepG2 cells. It is only in that (CaSR-activated) situation that Calhex 231, the CaSR negative allosteric modulator, prevented the secretion of the senescence-inducing factors into the CMcal. We did not expect CMcal to have an effect per se in this context.

  1. Regarding Figure 3. CMcal has a big effect on the Ki67 levels and seems to also downregulate its nuclear expression. How can this be explained, as an upregulation would be expected.

An increase could be expected if we had used calhex-231 directly on HepG2 cells. However, in our experiments we used conditioned media from cells that were treated with calhex. Therefore, HepG2 cells did not “see” the inhibitor, but rather just the effects that the inhibitor had preventing cinacalcet-induced preadipocyte secretion. Furthermore, the result shown in figure 3 is not significantly different from the CMveh condition.

  1. Regarding figure 4, the n=2 for the CMcal condition is underpowered and n-numbers should be increased. Furthermore, ELISA experiments should be performed to investigate whether HepG2 cells also secrete more cytokines. For this CM medium should be changed again to ensure the cytokines are deriving from HepG2 cells and not the SW872 cells. (Similarly, these experiments should be performed for Figure 11).

The n=2 is indeed a very small number, since the purpose of this condition was only to have an internal control (i.e., to confirm that the inhibitor did not have an effect per-se), and that is why we did not do more repetitions when we confirmed there was no effect. It was a mistake then to put it as one additional condition in the same level as the others, and we thank the reviewer for noticing. We now present this group in the graph as an inset, and more clearly explain that its purpose is to be an internal control. We have adapted the legends of all the figures having insets (figures 3-4, 7). On the other hand, to further characterize the senescent phenotype via ELISA is a good suggestion. However, we lack the material for this experiment, because we did not maintain the HepG2 cells for the required time to change the conditioned media for a fresh one to assess the suggested ELISA.

  1. This brings me to the next point, whether the observed effects are transient or long-lasting. If the CM is removed from the HepG2 cells, how long can the key observations then still be observed?

This is an excellent observation. In theory, the cellular senescence phenotype is irreversible. Since there usually is a heterogeneous response among cells in culture, if the conditioned media were removed, cells that initiated the senescent response would probably continue with self-perpetuating inflammatory process, which may affect a fraction of the other cells. In turn, those cells more resilient to the conditioned media may proliferate rapidly and form clusters, which would alter the conditions of the proposed model since the cells in the center of the clusters would not be exposed to the same environment, or would be stressed by the physical space. Thus, the reviewer’s question is important but difficult to address experimentally.

  1. Do the authors also observe other functional effects on HepG2 cells, like e.g. changes in lipid uptake capacity?

This is a good question, and we did not perform other functional analyses, as we intended to focus on the senescent phenotype and mitochondrial dysfunction. We added the issue in the discussion section as part of future research (lines 339-340).

  1. The authors describe an interesting phenomenon by which CaSR in pre-adipocytes induce senescence in author cells. Is this specific for hepatocytes or are comparable effects also seen in other cell types, like endothelial cells or smooth muscle cells. For this, selected key experiments should be replicated in one of these cell types.

This is a very good and important point. However, the suggested experiments exceed the original purpose of this manuscript. We included this suggestion of the possible effects on other metabolically relevant cell types in our discussion as a future research direction (lines 340-343).

Minor

  • Please double figure 7D, this seems to be the incorrect graph (and copy of figure 8D).

We apologize for this mistake; we have corrected the duplicate figure.

  • In Figure 9A all control values are 100. Please adjust the normalization so that also the variability in the control group can be assessed/visualized.

We improved the presentation on the graph by adjusting the normalization so the control group now has variability.

Reviewer 2 Report

This study is very interesting and has a scientific topic with a great impact on the field.

Minor comments:

Abstract and Introduction: These sections are well written and the aim of the study was clearly stated.

Materials and Methods: This section is ok, adequate and provided in detailed. 

Results: The results are very interesting but poorly presented. Figure 1 CMCIN should be changed to another in which the change is better seen. The authors should improve the quality of the images in figure 5.

Discussion: The authors could have combined Results and Discussion into a single section for a better explanation and understanding of the data provided.

References: This section is well written and up to date. Authors should review reference 30 and 33, they do not keep the format.

Author Response

Minor comments:

Abstract and Introduction: These sections are well written, and the aim of the study was clearly stated.

We thank the reviewer for his/her comment.

Materials and Methods: This section is ok, adequate and provided in detailed.

We thank the reviewer for his/her comment.

Results: The results are very interesting but poorly presented. Figure 1 CMcin should be changed to another in which the change is better seen. The authors should improve the quality of the images in figure 5.

According to the reviewer’s observation, we have improved the quality of the images for a better understanding of the results.

Discussion: The authors could have combined Results and Discussion into a single section for a better explanation and understanding of the data provided.

This is a good suggestion; however, we used the format of the template from the MDPI website, which styles the narrative in separate sections.

References: This section is well written and up to date. Authors should review reference 30 and 33, they do not keep the format.

According to the reviewer’s observation, we have corrected both references.

Reviewer 3 Report

In this manuscript, the authors have studied calcium sensing receptor (CaSR) in SW872 pre-adipocytes. Activation of CaSR increases the secretion of TNF-a and IL-1b. the authors investigated the impacts of CaSR-mediated secretion of these pro-inflammatory cytokines on senescence and mitochondrial dysfunction in hepatocytes HepG2. Conditioned medium from SW872 induced senescence and mitochondrial dysfunction when CaSR is activated. Overall, the study is interesting and promising. However, the authors need to address the following comments.

1.      The photos of SA-Beta Gal in Figure 1 are not clear.

2.      The effects of the CaRS should be confirmed by siRNA to specifically silence CaRS throughout the study.

3.      In Figure 2, the Western blots are too saturated making it hard to assess the effects of the treatments.

4.      If p53 is not upregulated, the authors should clarify the increased expression of p21 since it is p53-regulated.

5.      The authors should assess the levels of DNA damage upon treatments with CM.

6.      In figure 3, the authors should confirm the effect on cells cycle by FACS analysis.

7.      The changes in SASP in Figure 4 should be confirmed by ELISA. 

8.      In addition to immunoprecipitation of TNF-a (Figure 5), the authors should confirm its effect by silencing TNF-a in donor cells prior to the treatments with CM.

9.   In figure 5, other markers of senescence must be added to confirm the effects of TNF-a removal on senescence.

10.   Western blots in Figure 6 are too saturated and the changes are not clear.

11.   To confirm the dysfunctional mitochondria, the authors may investigate oxygen consumption rates.

12.   In Figure 11, the effects on SASP must be confirmed by ELISA.

Author Response

  1. The photos of SA-Beta Gal in Figure 1 are not clear.

According to the reviewer’s observation, we have improved the quality of the images for a better understanding of the results.

  1. The effects of the CaSR should be confirmed by siRNA to specifically silence CaSR throughout the study.

We agree with the reviewer that this is a limitation of our study. Unfortunately, the collection of conditioned media is a long and expensive process and thereby is our limiting reagent. We did not produce conditioned media from siRNA-treated SW872 cells because it would require a high number of cells, and the partial transfection efficiency might not provide conditioned media with the required quality. Thus, we preferred a pharmacological approach. Since this would have been an important contribution to the study, we have included it in the discussion as a limitation and suggestion for future research (lines 333-336).

  1. In Figure 2, the Western blots are too saturated making it hard to assess the effects of the treatments.

According to the reviewer’s observation, we have decreased the overall saturation of the images.

  1. If p53 is not upregulated, the authors should clarify the increased expression of p21 since it is p53-regulated.

This is an interesting point. We have included the following in the discussion hoping it will clarify the issue (lines 277-279): “Unlike p16 and p21, in our experiment we did not observe significant changes on p53, a protein that modulates p21 expression. In relation to this, it has been described that p21 can be regulated in a p53-independent manner through ERK1/2 and p38 MAPK (Hou et al., J Dermatol Sci 2022, 105, 88–97, doi:10.1016/J.JDERMSCI.2022.01.002).”

  1. The authors should assess the levels of DNA damage upon treatments with CM.

Indeed, the reviewer’s suggestion it is part of the methodologies to confirm the senescent phenotype. However, we did not perform said experiment, since we used up the available samples on the assays reported in the manuscript. We refer to this point in as a weakness of our research (lines 336-339).

  1. In figure 3, the authors should confirm the effect on cells cycle by FACS analysis.

We appreciate this suggestion and agree that it would be an excellent addition. However, because the conditioned media is our limiting reagent, we focused on a simpler, microscopy-based assay. We addressed the need to complement our findings with this method in the discussion section (lines 336-339).

  1. The changes in SASP in Figure 4 should be confirmed by ELISA.

We agree that this confirmation would enrich our work. However, we lack the material for this experiment, because we did not maintain the HepG2 cells for the required time to change the conditioned media for a fresh one to assess the suggested ELISA.

  1. In addition to immunoprecipitation of TNF-a (Figure 5), the authors should confirm its by silencing TNF-α in donor cells prior to the treatments with CM.

We agree with the reviewer that this is a limitation of our study. However, as indicated for query #2, conditioned media is our limiting reagent, and the production of conditioned from siRNA-treated SW872 cells might not have the required quality. Thus, we preferred the immunoprecipitation approach, which proved to be useful.

  1. In figure 5, other markers of senescence must be added to confirm the effects of TNF-α removal on senescence.

This is, indeed, a limitation of our study. Unfortunately, as indicated, the conditioned media is our limiting reagent and therefore focused on the single, most universal marker of cell senescence.

  1. Western blots in Figure 6 are too saturated and the changes are not clear.

According to the reviewer’s observation, we have decreased the overall saturation of the images.

  1. To confirm the dysfunctional mitochondria, the authors may investigate oxygen consumption rates.

We assessed oxygen consumption  (Figure 9), showing that CM from cinacalcet-treated SW872 cells decreases mitochondrial respiration. We added a short phrase (line 224) to clarify that this observation is indicative of mitochondrial dysfunction.

  1. In Figure 11, the effects on SASP must be confirmed by ELISA.

We agree that this confirmation would enrich our work. However, we lack the material for this experiment, because we did not maintain the HepG2 cells for the required time to change the conditioned media for a fresh one to assess the suggested ELISA.

Round 2

Reviewer 1 Report

I would like to thank the authors for there elaborate and appropriate response to my questions/remarks. I am satisfied and am happy to recommend this manuscript for publication.

Author Response

We thank the reviewer for his/her recommendation.

Reviewer 3 Report

The Authors have addressed all my comments for this paper and answered the technical questions I have for this method. 

Author Response

We thank the reviewer for his/her approval.